# Multi-source Domain Adaptation via Weighted Joint Distributions Optimal Transport

**Rosanna Turrisi**[1]        **Rémi Flamary**[2]        **Alain Rakotomamonjy**[3]        **Massimiliano Pontil**[4,5]

[1]DIBRIS, MaLGa, University of Genova, Genoa; CTSNC, Istituto Italiano di Tecnologia, Ferrara, Italy
[2]CMAP, École Polytechnique, Institut Polytechnique de Paris
[3]Criteo AI Lab, Paris
[4]CSML, Istituto Italiano di Tecnologia, Genoa, Italy
[5]Dept. of Computer Science, University College London, U.K.

## Abstract

This work addresses the problem of domain adaptation on an unlabeled *target* dataset using knowledge from multiple labelled *source* datasets. Most current approaches tackle this problem by searching for an embedding that is invariant across source and target domains, which corresponds to searching for a universal classifier that works well on all domains. In this paper, we address this problem from a new perspective: instead of crushing diversity of the *source* distributions, we exploit it to adapt better to the *target* distribution. Our method, named Multi-Source Domain Adaptation via Weighted Joint Distribution Optimal Transport (MSDA-WJDOT), aims at finding simultaneously an Optimal Transport-based alignment between the *source* and *target* distributions and a re-weighting of the *sources* distributions. We discuss the theoretical aspects of the method and propose a conceptually simple algorithm. Numerical experiments indicate that the proposed method achieves state-of-the-art performance on simulated and real datasets.

## 1 INTRODUCTION

Many machine learning algorithms assume that the test and training datasets are sampled from the same distribution. However, in many real-world applications, new data can exhibit a distribution change (*domain shift*) that degrades the algorithm performance. This shift can be observed for instance in computer vision when changing background, location, illumination or pose of the test images, or in speech recognition when the recording conditions or speaker accents are varying. To overcome this problem, Domain Adaptation (DA) [Jiang, 2008, Kouw and Loog, 2019] attempts to leverage labelled data from a *source* domain, in order to learn a classifier for unseen or unlabelled data in a *target* domain.

Several DA methods incorporate a distribution discrepancy loss into a neural network to overcome the domain gap. The distances between distributions are usually measured through an adversarial loss [Ganin et al., 2016, Ghifary et al., 2016, Tzeng et al., 2015, 2017] or integral probability metrics, such as the maximum mean discrepancy [Long et al., 2016, Tzeng et al., 2014]. DA techniques based on Optimal Transport have been proposed by [Courty et al., 2016, 2017, Damodaran et al., 2018] and justified theoretically by Redko et al. [2017].

In this work, we focus on the setting, more common in practice, in which several labelled *sources* are available, denoted in the following as multi-source domain adaptation (MSDA) problem. Many recent approaches motivated by theoretical considerations have been proposed for this problem. For instance, Mansour et al. [2009], Hoffman et al. [2018] provided theoretical guarantees on how several *source* predictors can be combined using proxy measures, such as the accuracy of a hypothesis. This approach can achieve a low error predictor on the *target* domain, under the assumption that the *target* distribution can be written as a convex combination of the *source* distributions.

Other MSDA methods [Peng et al., 2019, Zhao et al., 2018, Wen et al., 2020] look for a single hypothesis that minimizes the convex combination of its error on all *source* domains and they provide theoretical bounds of the error of the obtained hypothesis on the *target* domain. Those guarantees generally involve some terms depending on the distance between each *source* distribution and the *target* distribution and suggest to find an embedding in which the feature distributions between *sources* and *target* are as close as possible, by using Adversarial Learning [Zhao et al., 2018, Xu et al., 2018, Lin et al., 2020] or Moment Matching [Peng et al., 2019]. However, it may not be possible to find an embedding preserving discrimination even when the distances between *source* and *target* marginals are small. One such example is given in Figure 1, in which a rotation between the *sources*

*Accepted for the 38th Conference on Uncertainty in Artificial Intelligence* (UAI 2022).

prevents the existence of such invariant embedding as theorized by Zhao et al. [2019]. At last, we mention the very recent line of works on MSDA considering approaches inspired from imitation learning Nguyen et al. [2021a,b] and the work by Montesuma and Mboula [2021] building on Wasserstein barycenters.

**Contributions**  In this paper, we address the MSDA problem following a radically different route than the usual approach consisting in looking for a latent representation in which all *source* distributions are similar to the *target*. The approach we advocate embraces the diversity of *source* distributions and look for a convex combination of the joint *source* distributions with minimal Wasserstein distance to an estimated *target* distribution, without relying on a proxy measure such as the accuracy of *source* predictors.
We support this novel conceptual approach by deriving a generalization bound on the *target* error. Our algorithm consists in optimizing a key term in this generalization bound, given by the Wasserstein distance between the estimated joint *target* distribution and a weighted sum of the joint *source* distributions. One unique feature of our approach is that the weights of the *source* distribution are learned simultaneously with the classification function, which allows us to distribute the mass based on the similarity of the *sources* with the *target*, both in the feature and in the output spaces. As such, our model can also handle problems in which only target shift occurs. Interestingly the estimated weights provide a measure of domain relatedness and interpretability. We refer to the proposed method as Multi-Source Domain Adaptation via Weighted Joint Distribution Optimal Transport (MSDA-WJDOT).

**Notations**  Let $g : \mathcal{X} \to \mathcal{G}$ be a differentiable embedding function, with $\mathcal{G}$ the embedding space. Throughout the paper all input distributions are in this embedding space. We let $p_S$ and $p_T$ be the true joint distributions in the *source* and *target* domains, respectively. Both distributions are supported on the product space $\mathcal{G} \times \mathcal{Y}$, where $\mathcal{Y}$ is the label space. In practice we only have access to a finite number $N_S$ of samples in the *source* domain leading to the empirical *source* distribution $\hat{p}_S = \frac{1}{N_S} \sum_{i=1}^{N_S} \delta_{g(x_S^i), y_S^i}$ where $\delta$ is the Dirac function. In the *target* domain, only a finite number of unlabeled samples $N_T$ in the feature space is available. We then denote with $\Delta^J := \{\boldsymbol{\alpha} \in [0,1]^J \mid \sum_{i=1}^J \alpha_i = 1\}$ the $(J-1)$-dimensional simplex. Finally, given a loss function $L$ and a joint distribution $p$, the expected loss of a function $f$ is defined as $\varepsilon_p(f) = \mathbb{E}_{(x,y) \sim p}[L(y, f(x))]$.

## 2   OPTIMAL TRANSPORT AND DA

In this section we first recall the Optimal Transport problem and the notion of Wasserstein distance. Then we discuss how they were exploited for domain adaptation (DA) in the Joint Distribution Optimal Transport (JDOT) formulation that will be central in our approach.

**Optimal Transport**  The Optimal transport (OT) problem has been originally introduced by Monge [1781] and, reformulated as a relaxation by Kantorovich [1942]. Let $\hat{\mu}_S = \sum_i a_i^S \delta_{x_S^i}$, $\hat{\mu}_T = \sum_i a_i^T \delta_{x_T^i}$ be discrete probability measures with $\boldsymbol{a^S}, \boldsymbol{a^T} \in \Delta^J$. The OT problem searches a transport plan $\pi \in \Pi(\hat{\mu}_S, \hat{\mu}_T)$, where

$$\Pi(\hat{\mu}_S, \hat{\mu}_T) := \left\{ \pi \geq 0 \mid \sum_{i=1}^J \pi_{i,j} = a_j^T, \sum_{j=1}^J \pi_{i,j} = a_i^S \right\},$$

that is, the set of joint probabilities with marginals $\mu_1$ and $\mu_2$, that solve the following problem:

$$W_C(\hat{\mu}_S, \hat{\mu}_T) = \min_{\pi \in \Pi(\hat{\mu}_S, \hat{\mu}_T)} \sum_{i,j=1}^J C_{i,j} \cdot \pi_{i,j} \qquad (1)$$

where $C_{i,j} = c(x_S^i, x_T^j)$ represents the cost of transporting mass between $x_S^i$ and $x_T^j$ for a given ground cost function $c : \mathcal{X} \times \mathcal{X} \to \mathbb{R}_+$. It is often chosen to be the Euclidean distance, recovering the classical $W_1$ Wasserstein distance. Given a ground cost $C$, $W_C(\hat{\mu}_S, \hat{\mu}_T)$ corresponds to the minimal cost for mapping one distribution to the other and $\pi^\star$ is the OT matrix describing the relations between *source* and *target* samples. OT and in particular Wasserstein distance have been used with success in numerous machine learning applications such as Generative Adversarial Modeling [Arjovsky et al., 2017, Genevay et al., 2018] and DA [Courty et al., 2016, 2017, Shen et al., 2018].

**Joint Distribution Optimal Transport (JDOT)**  This method has been proposed by Courty et al. [2017] to address the problem of unsupervised DA with only one joint *source* distribution $\hat{p}_S$ and the feature marginal *target* distribution $\hat{\mu}_T$. Since no labels are available in the *target* domain, the authors proposed to use a proxy joint empirical distribution $\hat{p}_T^f$ whereby labels are replaced by the prediction of a classifier $f : \mathcal{G} \to \mathcal{Y}$, that is

$$\hat{p}_T^f = \frac{1}{N_T} \sum_{i=1}^{N_T} \delta_{g(x_T^i), f(g(x_T^i))}. \qquad (2)$$

In order to use a joint distribution in the Wasserstein distance, they defined, for $z, z' \in \mathcal{G}$ and $y, y' \in \mathcal{Y}$, the cost

$$D(z, y; z', y') = \beta \|z - z'\|^2 + L(y, y')$$

where $L$ is a loss between classes and $\beta$ weights the strength of feature loss. This cost takes into account embedding and label discrepancy. To train a meaningful classifier on the *target* domain, Courty et al. [2017] solved the problem

$$\min_f W_D(\hat{p}_S, \hat{p}_T^f) \qquad (3)$$

where the minimization is over a suitable set of classifiers and the objective $W_D(\hat{p}_S, \hat{p}_T^f)$ is a Wasserstein distance

between the joint *source* and joint "predicted" *target*,

$$\min_{\pi \in \Pi(\hat{p}_S, \hat{p}_T^f)} \sum_{i,j=1}^{J} D(g(x_S^i), y_S^i; g(x_T^j), f(g(x_T^j))) \cdot \pi_{i,j}.$$

JDOT has been supported by generalization error guarantees, [see Courty et al., 2017, for a discussion]. It was later extended to deep learning framework where the embedding $g$ was estimated simultaneously with the classifier $f$, via an efficient stochastic optimization procedure in [Damodaran et al., 2018]. A key aspect of JDOT, that was overlooked by the domain adaptation community, is the fact that the optimization problem involves the joint embedding/label distribution. This is in contrast to a large majority of DA approaches [Ganin et al., 2016, Sun and Saenko, 2016, Shen et al., 2018] using divergences only on the marginal distributions, whereas using simultaneously feature and labels information is the basis of most generalization bounds as discussed in the next section.

## 3  MULTI-SOURCE DA VIA WEIGHTED JOINT OPTIMAL TRANSPORT

We now discuss our MSDA approach. We assume to have $J$ *sources* with joint distributions $p_{S,j}$, for $1 \le j \le J$. We define a convex combination of the *source* distributions

$$p_S^\alpha = \sum_{j=1}^{J} \alpha_j p_{S,j} \tag{4}$$

with $\boldsymbol{\alpha} \in \Delta^J$ and we present a novel generalization bound for MSDA problem that depends on $p_S^\alpha$. Then, we introduce the MSDA-WJDOT optimization problem and propose an algorithm to solve it. Finally, we discuss the relation between MSDA-WJDOT and other MSDA approaches.

### 3.1  GENERALIZATION BOUND

The theoretical limits of DA are well studied and well understood since the work of Ben-David et al. [2010] that provided an "impossibility theorem" showing that, if the *target* distribution is too different from the *source* distribution, adaptation is not possible. However in the case of MSDA, one can exploit the diversity of the *source* domains and use only the *sources* close to the *target* distribution, thereby obtaining a better generalization bound. For this purpose, a relevant assumption, already considered in Mansour et al. [2009], is that the *target* distribution is a convex combination of the *source* distributions. The soundness of such an approach is illustrated by the following lemma.

**Lemma 1.** *For any hypothesis $f \in \mathcal{H}$, denote by $\varepsilon_{p_T}(f)$ and $\varepsilon_{p_S^\alpha}(f)$, the expected loss of $f$ on the target distribution*

*and on the weighted sum of the source distributions, with respect to a loss function $L$ bounded by $B$. Then*

$$\varepsilon_{p_T}(f) \le \varepsilon_{p_S^\alpha}(f) + B \cdot D_{TV}(p_S^\alpha, p_T) \tag{5}$$

*where $D_{TV}$ is the total variation distance.*

This simple inequality, whose proof is presented in the appendix, tells us that the key point for *target* generalization is to have a function $f$ with low error on a combination of the joint *source* distributions and that combination should be "near" to the *target* distribution. Note that this also holds for single *source* DA problem corroborating the recent findings that just matching marginal distributions may not be sufficient [Wu et al., 2019]. While the above lemma provides a simple and principled guidance for a multi-source DA algorithm, it cannot be used for training since it assumes that labels in the *target* domain are known. In the following, we provide a generalization bound in a realistic scenario where no *target* labels are available and a self-labelling strategy is employed to compensate for the missing labels.

Taking inspiration from the result in Lemma 1, we propose a theoretically grounded framework for learning from multiple *sources*. To this end, we first recall the notion of Probabilistic Transfer Lipschitzness (PTL) of a classifier Courty et al. [2017], that will be used in our method.

**Definition 1.** (PTL Property) *Let $D$ be a metric on $\mathcal{G}$ and let $\phi : \mathbb{R} \to [0, 1]$. A labeling function $f : \mathcal{G} \to \mathbb{R}$ and a joint distribution $\pi \in \Pi(\mu_S, \mu_T)$ are $\phi$-Lipschitz transferable if for all $\lambda > 0$, we have*

$$\mathrm{Prob}_{(x_S, x_T) \sim \pi}\big[|f(x_S) - f(x_T)| > \lambda D(x_S, x_T)\big] \le \phi(\lambda).$$

The PTL property is a reasonable assumption for DA that was introduced in Courty et al. [2017] and provides a bound on the probability of finding pair of *source-target* samples of different label within a $1/\lambda$-ball.

Our approach is based on the idea that one can compensate the lack of *target* labels by using an hypothesis labelling function $f$ which provides a joint distribution $p_T^f$ in (2), where $f$ is searched in order to align $p_T^f$ with a weighted combination of *source* distributions $p_S^\alpha$. Following this idea, we introduce the definition of similarity measure and a new generalization bound for MSDA.

**Definition 2.** (Similarity measure) *Let $\mathcal{H}$ be a space of $M$-Lipschitz labelling functions. Assume that, for every $f \in \mathcal{H}$ and $x, x' \in \mathcal{G}$, $|f(x) - f(x')| \le M$. Consider the following measure of similarity between $p_S^\alpha$ and $p_T$ introduced in [Ben-David et al., 2010, Def. 5]*

$$\Lambda(p_S^\alpha, p_T) = \min_{f \in \mathcal{H}} \varepsilon_{p_S^\alpha}(f) + \varepsilon_{p_T}(f), \tag{6}$$

*where the risk is measured w.r.t. to a symmetric and k-Lipschitz loss function that satisfies the triangle inequality.*

**Theorem 1.** *Let $\mathcal{H}$ be the space introduced in Definition 2 and assume that the function $f^*$ minimizing Eq. 6 satisfies the PTL property (Definition 1). Let $\hat{p}_{S,j}$ be $j$-th source empirical distributions of $N_j$ samples and $\hat{p}_T$ the empirical target distribution with $N_T$ samples. Then for all $\lambda > 0$, with $\beta = \lambda k$ in the ground metric $D$, we have with probability at least $1 - \eta$ that*

$$\varepsilon_{p_T}(f) \leq W_D\left(\hat{p}_S^{\boldsymbol{\alpha}}, \hat{p}_T^f\right) + \sqrt{\frac{2}{c'}\log\frac{2}{\eta}}\left(\frac{1}{N_T} + \sum_{j=1}^{J}\frac{\alpha_j}{N_j}\right)$$
$$+ \Lambda(p_S^{\boldsymbol{\alpha}}, p_T) + kM\phi(\lambda).$$

Note that the quantity $\Lambda(p_S^{\boldsymbol{\alpha}}, p_T)$ in the bound measures the discrepancy between the true *target* distribution and the "best" combination of the *source* distributions and, similarly to some terms in the DA bounds of Ben-David et al. [2010], it is not directly controllable. However, we have experimentally checked that our approach minimizes an upper bound of this term $\Lambda$ – see discussion in Section 4 and Figure **??** in the appendix. Interestingly the $1/N_j$ ratios in the bound are weighted by $\alpha_j$ which means that even if one *source* is poorly sampled it won't have a large impact as soon as the coefficient $\alpha_j$ stays small. This suggests to investigate some kind of regularization for the weights $\boldsymbol{\alpha}$ but since it would introduce one more hyperparameter we left it to future works and in the following focus only on optimizing the first term of the bound.

### 3.2 MSDA-WJDOT PROBLEM

**MSDA-WJDOT Optimization Problem** Our approach aims at finding a function $f$ that aligns the distribution $p_T^f$ with a convex combination $\sum_{j=1}^{J}\alpha_j p_{S,j}$ of the *source* distributions with convex weights $\boldsymbol{\alpha} \in \Delta^J$ on the simplex. We express the multi-domain adaptation problem as

$$\min_{\boldsymbol{\alpha},f}\quad W_D\left(\hat{p}_T^f, \sum_{j=1}^{J}\alpha_j\hat{p}_{S,j}\right). \tag{7}$$

Problem above is a minimization of the first term in the bound from Theorem 1 with respect to both $f$ and $\boldsymbol{\alpha}$. The role of the weight $\boldsymbol{\alpha}$ is crucial because it allows in practice to select (when $\boldsymbol{\alpha}$ is sparse) the *source* distributions that are the closest in the Wasserstein sense and use only those distributions to transfer label knowledge from. An example of the method is provided in Figure 1 showing 4 *source* distributions in 2D obtained from rotation in the 2D space. One interesting property of our approach is that it can adapt to a lot of variability in the *source* distributions as long as the distributions lie in a distribution manifold and this

**Algorithm 1** Optimization for MSDA-WJDOT

---

Initialise $\boldsymbol{\alpha} = \frac{1}{J}\mathbf{1}_J$ and $\boldsymbol{\theta}$ parameters of $f_{\boldsymbol{\theta}}$ and steps $\mu_{\boldsymbol{\alpha}}$ and $\mu_{\boldsymbol{\theta}}$.
**repeat**
    $\boldsymbol{\theta} \leftarrow \boldsymbol{\theta} - \mu_{\boldsymbol{\theta}}\nabla_{\boldsymbol{\theta}}W_D\left(\hat{p}_T^f, \sum_{j=1}^{J}\alpha_j\hat{p}_{S,j}\right)$
    $\boldsymbol{\alpha} \leftarrow P_{\Delta^J}\left(\boldsymbol{\alpha} - \mu_{\boldsymbol{\alpha}}\nabla_{\boldsymbol{\alpha}}W_D(\hat{p}_T^f, \sum_{j=1}^{J}\alpha_j\hat{p}_{S,j})\right)$
**until** Convergence

---

manifold is sampled correctly by the *source* distributions. For instance the linear weights allow to interpolate between *source* distributions and recover the weighted *source* that is the closest to the manifold of distribution, hence providing a tightest generalization as shown in the previous section.

**Optimization Algorithm** Problem (7) can be solved with a block coordinate descent similarly to what was proposed in Courty et al. [2017]. But with the introduction of the weights $\boldsymbol{\alpha}$ we numerically observed that one can easily get stuck in a local minimum with poor performances. So we proposed the optimization approach in Algorithm 1, that is an alternated projected gradient descent *w.r.t.* the parameters $\boldsymbol{\theta}$ of the classifier $f_{\boldsymbol{\theta}}$ and the weights $\boldsymbol{\alpha}$ of the sources. Note that the sub-gradient of $\nabla_{\boldsymbol{\theta}}W$ is computed by solving the OT problem and using the fixed OT matrix to compute the gradient similarly to Damodaran et al. [2018]. It is well known that the subgradient *w.r.t.* the weights of a distribution can be expressed as $\nabla_{\boldsymbol{w}}W(\mu, \sum_{i=1}^{J}w_i\delta_{x_i}) = \boldsymbol{\beta}$ where $\boldsymbol{\beta}$ is the optimal right dual variable of the problem. Moreover, the sub-gradient $\nabla_{\boldsymbol{\alpha}}W$ can be computed in closed form as

$$\nabla_{\alpha_j}W_D\left(\hat{p}_T^f, \sum_{j=1}^{J}\frac{\alpha_j}{N_j}\sum_{i=1}^{N_j}\delta_{(g(x_j^i),y_j^i)}\right) = N_j\sum_{i=1}^{N_j}\beta_{j,i}^*$$

where $\beta_{j,i}^*$ is the dual variable for sample $i$ in source domain $j$. The definition of the projection to the simplex $P_\Theta$ is provided in supplementary materials. Also note that while we did not need it in the numerical experiments, Algorithm 1 can be performed on mini-batches by sub-sampling the *source* and *target* distribution on very large datasets as suggested in Damodaran et al. [2018] which has been shown to provide robust estimators in Fatras et al. [2020].

### 3.3 RELATED WORK

**MSDA approaches learning only the classifier** MSDA-WJDOT is related to JDOT [Courty et al., 2017] but proposes a non-trivial extension of it to multisource domain adaption. Indeed, there are two simple ways to apply JDOT to multi-source DA, which we refer to as Concatenated JDOT (CJDOT) and Multiple JDOT (MJDOT). The first one consists in concatenating all the *source* samples into one *source* distribution (equivalent to uniform $\boldsymbol{\alpha}$ if all $N_j$ are

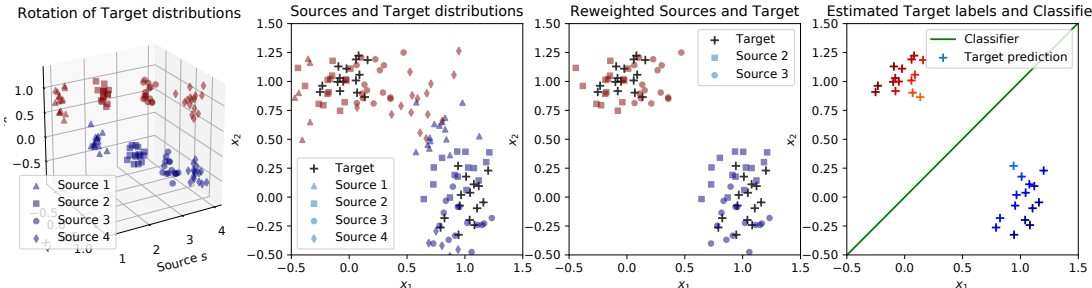

Figure 1: 2D simulated data. (Left) illustration of 4 *source* distributions corresponding to 4 increasing rotations. The color of the sample corresponds to the class. (Center Left) *source* distributions and *target* distribution in black because no class information is available. (Center Right) *source* distributions weighted by the optimal $\boldsymbol{\alpha}^\star = [0, 0.5, 0.5, 0]$ from MSDA-WJDOT: only Source 2 and 3 have a weight $> 0$ because they are the closest to the *target* in the Wasserstein sense. (Right) Final MSDA-WJDOT *target* classification.

equal) and using classical JDOT on the resulting distribution. The second one consists in optimizing a sum of JDOT losses for every *source* distribution but again, this leads to uniform impact of the *sources* on the estimation. It is clear that both approaches are not robust when some *sources* distributions are very different from the *target* (those would have a small weight in MSDA-WJDOT). Recently, Montesuma and Mboula [2021] proposed to compute a Wasserstein barycenter to aggregate the source marginal distributions. Once the intermediate domain is computed, they transport the Wasserstein barycenter into the target domain using the Sinkhorn algorithm [Cuturi, 2013] with (WTB$_{reg}$) or without (WTB) class regularization. The Wasserstein barycenter is also used in another MSDA approach, called JCPOT [Redko et al., 2019], to estimate the class proportion. This method, based on Courty et al. [2016], has been proposed to address only *target* shift (change in proportions between the classes) and satisfies a generalization bound showing that estimating the class proportion in the *target* distribution is key to recovering good performances. MSDA-WJDOT can also handle the *target* shift as a special case since the reweighting $\boldsymbol{\alpha}$ is directly related to the proportion of classes. A crucial difference between MSDA-WJDOT and the barycenter-based approaches described above is that they rely only on aligning marginal distributions, whereas the proposed method aligns joint distributions by optimizing a Wasserstein distance in the joint embedding/label space.

Also note that MSDA-WJDOT relies on a weighting of the samples where the weight is shared inside the *source* domains. This is a similar approach to DA approaches such as Importance Weighted Empirical Risk Minimization (IWERM) [Sugiyama et al., 2007] designed for Covariate Shift that use a reweighing of all the samples. One major difference is that we only estimate a relatively small number of weights in $\boldsymbol{\alpha}$ leading to a better posed statistical estimation. It is indeed well known that estimation of continuous density which is necessary for a proper individual reweighting of the samples is a very difficult problem in high dimension. All the above mentioned methods do not require to

learn an embedding, whose estimation may be computationally expensive and unnecessary (e.g., when a pre-trained model is available). Further, there exists numerous examples of *source* variability in real life (such as rotation between the full distributions) that cannot be handled with a global embedding.

**MSDA approaches estimating an embedding** As discussed in the introduction, the majority of recent DA approaches based on deep learning [Ganin et al., 2016, Sun and Saenko, 2016, Shen et al., 2018] relies on the estimation of an embedding that is invariant to the domain which means that the final classifier is shared across all domains when the embedding $g$ is estimated. Those approaches have been extended to multiple *sources* with the objective that the embedded distributions between *sources* and *target* are similar. Authors in Xu et al. [2018] propose an algorithm based on adversarial learning, named Deep Cocktail Network (DCTN), to learn a feature extractor, domain discriminators and *source* classifiers. The domain discriminator provides multiple source-target-specific perplexity scores that are used to weight the source-specific classifier predictions and produce the *target* estimation. In Peng et al. [2019], the embedding is learned by aligning moments of the *source* and *target* distributions, by an approach called Moment matching (M$^3$SDA) . Our approach differs greatly here as we do not try to cancel the variability across *sources* but to embrace it by allowing the approach to automatically find the *source* domains closest in terms of embedding and labeling function.

# 4 NUMERICAL EXPERIMENTS

In this section, we first discuss the implementation and the robustness of MSDA-WJDOT. We then evaluate and compare it with state-of-the-art MSDA methods, on both simulated and real data. The numerical implementation relies on the Pytorch [Paszke et al., 2017] and Python Optimal Trans-

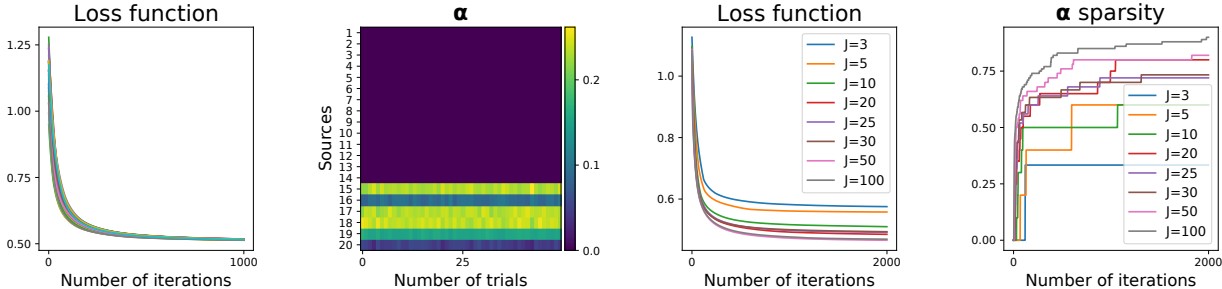

Figure 2: (Left and Center-Left) Loss function and $\alpha$ coefficients with different weights initializations. (Center-Right and Right) Loss function and $\alpha$ sparsity for increasing number of sources $J$.

port [Flamary et al., 2021] toolboxes and will be released upon publication.

**Practical Implementation** We used in all numerical experiments the MSDA-WJDOT solver from Algorithm 1. We recall that in this paper we assume to have access to a meaningful (as in discriminant) embedding $g$. This is a realistic scenario due to the wide availability of pre-trained models and advent of reproducible research. Nevertheless we discuss here how to estimate such an embedding when none is available. To keep the variability of the *sources* that is used by MSDA-WJDOT we propose to estimate $g$ with the Multi-Task Learning framework originally proposed in Caruana [1997], i.e.

$$\min_{g, \{f_j\}_{j=1}^J} \quad \sum_{j=1}^J \frac{1}{N_j} \sum_{i=1}^{N_j} \mathcal{L}(f_j \circ g(x_j^i), y_j^i). \quad (8)$$

This approach for estimating an embedding $g$ makes sense because it promotes a $g$ that is discriminant for all tasks but allows a variability thanks to the task specific final classifiers $f_j$ which is an assumption at the core of MSDA-WJDOT. We refer to MSDA-WJDOT where the embedding $g$ is learned with the above procedure as MSDA-WJDOT$_{MTL}$. Note that this is a two step procedure.

An important question, especially when performing unsupervised DA, is how to perform the validation of the parameters including early stopping. We propose here to use the sum of squared errors (SSE) between the *target* points in the embedding and their cluster centroids. Specifically, we estimate cluster membership on the the outputs through $f \circ g$. Then the SSE is computed in the embedding $g$ using the estimated clusters. Intuitively, if the SSE decreases it means that $f$ attributes the same label to samples of the target domain that are close in the embedding. We also explored another strategy, based on the classifier accuracy on the sources, that is discussed and reported in the supplementary material.

In addition, to provide a lower and an upper bound of the MSDA performance, we implemented supervised classification methods trained on the *sources* (`Baseline`), the *target* (`Target`), on both *sources* and *target* (`Baseline+Target`) domain. We consider `Baseline` as a performance lower bound as the *target* domain is not used during training, whereas `Target` and

`Baseline+Target` are two unrealistic approaches that use labels in *target*. Note that `Target` trains a classifier using only *target* labels and is more prone to overfitting since less samples are available. Since we have access to labels for `Target` and `Baseline+Target`, we validate the model by using the classification accuracy on the *target* validation set making those two approaches clear upper bounds on the attainable performance for each dataset. All methods are compared on the same dataset split in training (70%), validation (20%) and testing (10%) but the validation set is used only for `Baseline+Target` and `Target`.

**Algorithm convergence and stability** In Figure 2 (*Left* and *Center-left*) we show the stability of the algorithm for different weights initialization. The loss function always converges and the $\boldsymbol{\alpha}$ coefficients are not affected by the initialization. Moreover, we observed in practice that choosing the same step for $\boldsymbol{\alpha}$ and $\boldsymbol{\theta}$ does not degrade the performance and in all experiments we validated it via early stopping. We also noticed a fast convergence of the weights $\boldsymbol{\alpha}$, meaning that the relevant domains are quickly identified. This behavior is illustrated in Fig. 2 (*Right*), where $\boldsymbol{\alpha}$ sparsity rapidly increases for any choice of $S$ illustrating that only few relevant source distributions are used in practice. We also report the loss convergence for increasing number of sources $S$ (*Center-right*).

**Simulated Data: Domain Shift** We consider a classification problem similar to what is illustrated in Figure 1, but with 3 classes, i.e. $\mathcal{Y} = \{0, 1, 2\}$, and in 3D. For the *sources* and *target* we generate $N_j$ and $N_T$ samples from $J + 1$ Gaussian distributions rotated of angle $\theta_j \in [0, \frac{3}{2}\pi]$ around the $x$-axis. As the data is already linearly separated, we set $g$ as the identity function in this experiment. We carried out many experiments in order to see the effect of different parameters such as the number of *source* domains $J$, of *source* samples $N_j$ and of *target* samples $N_T$. Each experiment has been repeated 50 times. We report in Fig. 3 the accuracy of all methods with $N_j = N_T = 300$ for $J = 3$ (Left) and $J = 30$ (Right). All competing methods are clearly outperformed by MSDA-WJDOT both in term of performance and variance even for a limited number of sources. Interestingly MSDA-WJDOT can even outperform `Target` due to its

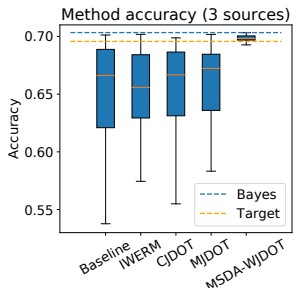
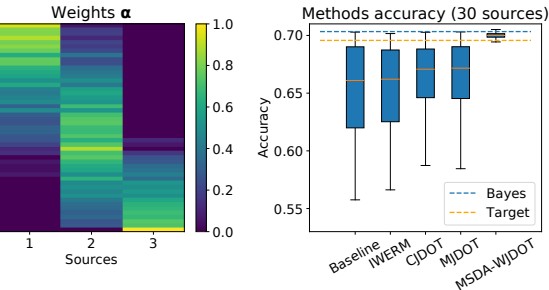
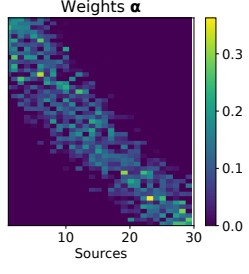

Figure 3: Simulated dataset. Methods' accuracy and recovered $\alpha$ weights for an increasing rotation angle of the *target* samples: (Left and Center-Left) $J = 3$ and (Right and Center-Right) $J = 30$ sources.

access to a larger number of samples. Another important aspect of MSDA-WJDOT is the obtained weights $\alpha$ that can be used for interpretation. We show in Fig. 3 the $\alpha$ weights that are attributed to the *sources* (ordered on the $x$-axis by increasing rotation angles), for in an increasing rotation angle in the *target* samples ($y$-axis). The estimated weights tend to be sparse and put more mass on *sources* that have a similar angle *i.e.* we recover automatically the closest *sources* in the joint distribution manifold. Note that we only report the method's performances on those two configurations; the results for other experiments can be found in the supplementary material.

We next investigate how the function $\Lambda$ in (6) behaves when the weights $\alpha$ are optimized *w.r.t.* the first term of the bound in Theorem 1. To this end we computed for 30 sources an upper bound of $\Lambda$ with the 0-1 loss by using the estimated $\hat{f}$ instead of the minimizer in (6). We recover a value of $0.57$ that is very close to twice the Bayes error, corresponding to the best possible value for $\Lambda$ in this experiment. On the other hand, the value for the upper bound of $\Lambda$ for a uniform $\alpha$ is $0.64$ and $0.65$ in average for 10000 randomly drawn values of $\alpha$. This suggests that optimizing $\alpha$ with MSDA-WJDOT leads to a minimization of $\Lambda$ in the generalization bound.

**Simulated Data: Target Shift** We take into account the target shift problem with 2D *source* and *target* datasets which present different proportions of classes. The proportion of the class $c$ in the *source* $j$ is defined as $P_j^c = \frac{\#\{y_j^i = c\}}{N_j}$ (and similarly for the *target*). We consider a binary classification task and we sample *sources* and *target* datasets from the same Gaussian distribution. In Fig. 4 (Left and Center) we illustrate two *sources* and *target* distributions and how MSDA-WJDOT reweights the *sources*. As we can see, almost all the mass is concentrated on Source 2 ($\alpha_2 \gg \alpha_1$) because its class proportion is closer to the *target* one. Instead, Source 1 has a class proportion inverted w.r.t. the *target*. In the experiment reported in Fig. 4 (Center-Right and Right) we have $J = 20$ *sources* with $P_j^2$ randomly generated between $0.1$ and $0.9$ (we ordered the *sources* s.t. $P_j^2 \leq P_{j+1}^2$). We show the average classification accuracy and the $\alpha$ weights over 50 trials for varying $P_T^2$ in $\{0.1, 0.2, \cdots, 0.9\}$. Our method always outperforms JCPOT and selects the *sources* with a proportion of classes closer to the one in the *target*.

**Object Recognition** The Caltech-Office dataset [Gong et al., 2012] contains four different domains: Amazon, Caltech [Griffin et al., 2007], Webcam and DSLR. The variability of the different domains come from several factors: presence/absence of background, lightning conditions, noise, etc. We use for the embedding function $g$ the output of the 7th layer of a pre-trained DeCAF model [Donahue et al., 2014], similarly to what was done in Courty et al. [2016], resulting into an embedding space $\mathcal{G} \in \mathbb{R}^{4096}$. For $f$, we employ a one-layer neural network. Training is performed with Adam optimizer with 0.9 momentum and $\epsilon = e^{-8}$. Learning rate and $\ell_2$ regularization on the parameters are validated for all methods. In JDOT extensions and MSDA-WJDOT, we also validate the $\beta$ parameter weighting the feature distance in the cost (3).

The aim of this experiment is to evaluate MSDA-WJDOT and compare it with the current literature in the setting in which the embedding is given. The performance of the methods is reported in Table 1. We can see that MSDA-WJDOT is state of the art providing the best Average Rank (AR). Note that the DeCAF pre-trained embedding was originally designed in part to minimize the divergence across domains which as discussed is not the best configuration for MSDA-WJDOT but it still performs very well showing the robustness of MSDA-WJDOT to the embedding. Moreover, we observed that for each adaptation problem MSDA-WJDOT provides one-hot vector $\alpha$ (reported in supplementary) suggesting that only one *source* is needed for the *target* adaptation. Interestingly the source selected by MSDA-WJDOT for each target is the one that was reported with the best performance for single-source DA in Courty et al. [2016], which shows that MSDA-WJDOT can automatically find the relevant sources with no supervision.

**Music-speech Discrimination** We now tackle a MSDA problem in which both the embedding and the *target* classifier need to be learned. Specifically, we consider the music-speech discrimination task introduced in Tzanetakis and Cook [2002], which includes 64 music and speech tracks of 30 seconds each. We generated 14 noisy datasets by combining the raw tracks with different types of noises from a noise

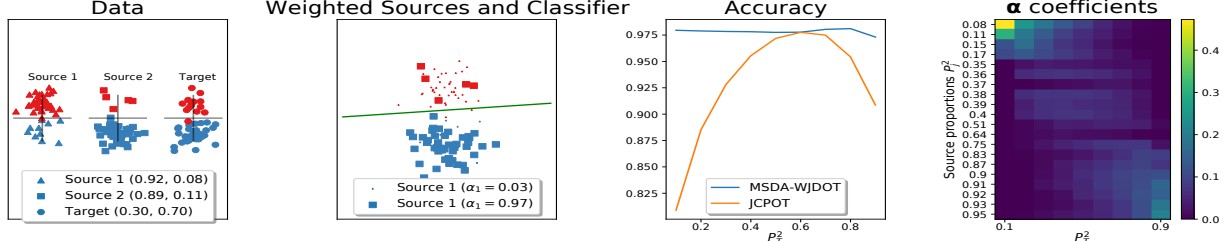

Figure 4: Illustration of MSDA-WJDOT on target shift problem. (Left) illustration of 2 *source* and *target* distributions with unbalanced classes. (Center-Left) *source* distributions weighted by $\alpha$ and estimated target classifier. (Center-Right and Right) classification accuracy of MSDA-JDOT and JCPOT and $\alpha$ coefficients at varying of class proportions in *target* dataset.

Table 1: Object recognition accuracy. The last column reports the average rank across target domain. Results of methods marked by $*$ are from Montesuma and Mboula [2021].

| Method | Amazon | dslr | webcam | Caltech10 | AR |
|---|---|---|---|---|---|
| Baseline | $93.13 \pm 0.07$ | $94.12 \pm 0.00$ | $89.33 \pm 1.63$ | $82.65 \pm 1.84$ | 5.00 |
| IWERM | $93.30 \pm 0.75$ | $\mathbf{100.00 \pm 0.00}$ | $89.33 \pm 1.16$ | $\mathbf{91.19 \pm 2.57}$ | 2.75 |
| CJDOT | $93.71 \pm 1.57$ | $93.53 \pm 4.59$ | $90.33 \pm 2.13$ | $85.84 \pm 1.73$ | 3.50 |
| MJDOT | $94.12 \pm 1.57$ | $97.65 \pm 2.88$ | $90.27 \pm 2.48$ | $84.72 \pm 1.73$ | 3.00 |
| JCPOT$*$ | $79.23 \pm 3.09$ | $81.77 \pm 2.81$ | $93.93 \pm 0.60$ | $77.91 \pm 0.45$ | 5.50 |
| WBT$*$ | $59.86 \pm 2.48$ | $60.99 \pm 2.15$ | $64.13 \pm 2.38$ | $62.80 \pm 1.61$ | 7.25 |
| WBT$*_{reg}$ | $92.74 \pm 0.45$ | $95.87 \pm 1.43$ | $\mathbf{96.57 \pm 1.76}$ | $85.01 \pm 0.84$ | 4.00 |
| MSDA-WJDOT | $\mathbf{94.23 \pm 0.90}$ | $\mathbf{100.00 \pm 0.00}$ | $89.33 \pm 2.91$ | $85.93 \pm 2.07$ | $\mathbf{2.25}$ |
| Target | $95.77 \pm 0.31$ | $88.35 \pm 2.76$ | $99.87 \pm 0.65$ | $89.75 \pm 0.85$ | - |
| Baseline+Target | $94.78 \pm 0.48$ | $99.88 \pm 0.82$ | $100.00 \pm 0.00$ | $91.89 \pm 0.69$ | - |

Table 2: Music-Speech discrimination accuracy and average rank across target domains. Results of methods marked by $*$ are from Montesuma and Mboula [2021].

| Method | F16 | B2 | F2 | D | AR |
|---|---|---|---|---|---|
| Baseline | $69.67 \pm 8.78$ | $57.33 \pm 7.57$ | $83.33 \pm 9.13$ | $87.33 \pm 6.72$ | 9.25 |
| IWERM | $72.22 \pm 3.93$ | $58.33 \pm 5.89$ | $85.00 \pm 6.23$ | $81.64 \pm 3.33$ | 8.75 |
| IWERM$_{MTL}$ | $75.00 \pm 0.00$ | $66.67 \pm 0.00$ | $\mathbf{100.00 \pm 0.00}$ | $98.33 \pm 3.33$ | 4.00 |
| DCTN | $66.67 \pm 3.61$ | $68.75 \pm 3.61$ | $87.50 \pm 12.5$ | $94.44 \pm 7.86$ | 6.50 |
| M$^3$SDA | $70.00 \pm 4.08$ | $61.67 \pm 4.08$ | $85.00 \pm 11.05$ | $83.33 \pm 0.00$ | 8.50 |
| CJDOT | $59.50 \pm 13.95$ | $50.00 \pm 0.00$ | $83.33 \pm 0.00$ | $91.67 \pm 0.00$ | 9.75 |
| CJDOT$_{MTL}$ | $83.83 \pm 5.11$ | $74.83 \pm 1.17$ | $\mathbf{100.00 \pm 0.00}$ | $95.74 \pm 16.92$ | 3.25 |
| MJDOT | $66.33 \pm 9.57$ | $50.00 \pm 0.00$ | $83.33 \pm 0.00$ | $91.67 \pm 0.00$ | 9.50 |
| MJDOT$_{MTL}$ | $86.00 \pm 4.55$ | $72.83 \pm 5.73$ | $97.67 \pm 3.74$ | $97.74 \pm 8.28$ | 3.50 |
| JCPOT$*$ | $88.67 \pm 1.67$ | $92.55 \pm 2.11$ | $82.41 \pm 2.22$ | $87.89 \pm 1.39$ | 5.50 |
| WBT$*$ | $56.63 \pm 6.56$ | $56.88 \pm 9.54$ | $59.38 \pm 2.61$ | $56.63 \pm 6.88$ | 11.75 |
| WBT$*_{reg}$ | $\mathbf{94.92 \pm 0.68}$ | $\mathbf{96.27 \pm 1.60}$ | $96.87 \pm 0.94$ | $92.98 \pm 1.38$ | 3.00 |
| MSDA-WJDOT | $83.33 \pm 0.00$ | $58.33 \pm 6.01$ | $87.00 \pm 6.05$ | $89.00 \pm 4.84$ | 7.00 |
| MSDA-WJDOT$_{MTL}$ | $87.17 \pm 4.15$ | $74.83 \pm 1.20$ | $99.67 \pm 1.63$ | $\mathbf{99.67 \pm 1.63}$ | $\mathbf{2.25}$ |
| Target | $73.67 \pm 6.09$ | $69.17 \pm 7.50$ | $77.33 \pm 4.73$ | $73.17 \pm 9.90$ | - |
| Baseline+Target | $71.06 \pm 9.31$ | $67.62 \pm 11.92$ | $85.33 \pm 11.85$ | $79.53 \pm 10.05$ | - |

dataset (`spib.linse.ufsc.br/noise.html`). The noisy datasets have been synthesised by PyDub python library [Robert et al., 2018]. We then used the libROSA python library [Brian McFee et al., 2015] to extract 13 MFCCs, computed every 10ms from 25ms Hamming windows followed by a z-normalization per track. We chose each of the four noisy datasets F16, Bucaneer2 (B2), Factory2 (F2), and Destroyerengine (D) as *target* domains, considering the remaining noisy datasets and the clean dataset as labelled *source* domains. The feature extraction $g$ is a Bidirectional Long Short-Term Memory (BLSTM) recurrent network with 2 hidden layers of 50 memory blocks each. The $f$ classifier is learned as one feed-forward layer. Model and training details are reported in the supplementary materials.

We report in Table 2, the mean and standard deviation accuracy on the testing set of each *target* dataset over 50 trials, as well as the Average Rank for each method. First note that on this hard adaptation problem the `Baseline+Target` approach only slightly improves the `Baseline`, and most of the methods performance shows large variance. As expected, MSDA-WJDOT$_{MTL}$ significantly outperforms MSDA-WJDOT confirming the importance of estimating an embedding $g$ exploiting the *source* variability. MSDA-WJDOT$_{MTL}$ achieves a 1.25 Average Rank outperforming all the other MSDA methods and also presents low standard deviation, showing robustness to small sample size. Surprisingly, MSDA-WJDOT$_{MTL}$ even outper-

forms both the `Target` and `Baseline+Target` methods, where the labels are available.

## 5 CONCLUSION

We presented a novel approach for multi-source DA that relies on OT for propagating labels from the *sources* and a weighting of the *source* domains that selects the best *sources* for the *target* task at hand in order to get a better prediction. We provided results that show that the proposed approach is theoretically grounded. We present numerical experiments on simulated data that shows the effectiveness of our method on both *domain* and *target shift* problems. Finally, we illustrate the good performance of MSDA-WJDOT on real-world benchmark datasets. Future works will investigate a regularization of $\alpha$ and estimating simultaneously the embedding $g$ with MSDA-WJDOT instead of pre-training it with multitask learning. The embedding could indeed be updated for each new *target* which suggests an incremental formulation for MSDA-WJDOT that could be valuable in practice.

## Acknowledgements

This work was partially funded through the 3IA Cote d'Azur Investments ANR-19-P3IA-0002 of the French National Research Agency (ANR), the DECIPHER-ASL – Bando PRIN 2017 grant (2017SNW5MB - Ministry of University and Research, Italy), and a grant from SAP SE and 5x1000, assigned to the University of Ferrara - tax return 2017. This research was produced within the framework of Energy4Climate Interdisciplinary Center (E4C) of IP Paris and Ecole des Ponts ParisTech. This research was supported by 3rd Programme d'Investissements d'Avenir ANR-18-EUR-0006-02. This action benefited from the support of the Chair "Challenging Technology for Responsible Energy" led by l'X – Ecole polytechnique and the Fondation de l'Ecole polytechnique, sponsored by TOTAL.

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
