# OpenReview forum: "Multi-source Domain Adaptation via Weighted Joint Distributions Optimal Transport"
_auai.org/UAI/2022/Conference — UAI 2022 Poster_

### Official Review · Reviewer_RT94 · 2022-03-30

**Q2(1) Originality/Novelty:** 3
**Q2(2) Significance/Impact:** 3
**Q2(3) Correctness/Technical Quality:** 3
**Q2(6) Clarity Of Writing:** 4
**Q6 Overall Score:** 7
**Q8 Confidence In Your Score:** 3

**Q1 Summary And Contributions:**

The paper considers the setting under which multiple labelled sources are available and a target source without lables must be treated.
Main contributions
1) builds latent representations in which all source distributions are similar to the target
2) provides a theoretical bound on generalization for the target error
3) weights of the sources' distribution are learned jointly with the classification function, thus allowing to distribute the mass based on the sources-target similarity.

**Q2 Assessment Of The Paper:**

More detailed information regarding each of these aspects is given below:

**Q2(4) Quality Of Experiments (Optional):**

3: Good: The experimental evaluation is adequate, and the results convincingly support the main claims.

**Q2(5) Reproducibility:**

3: Good: Key resources (e.g., proofs, code, data) are available and key details (e.g., proofs, experimental setup) are sufficiently well-described for competent researchers to confidently reproduce the main results.

**Q3 Main Strengths:**

The main strenghts of the paper are the following:
1) the tackled problem is a relavant one and it is tackled in a new and fresh way when compared to existing approaches from the specialized literature.
2) the theoretical analysis, which provides bound on generalization over the target source, is extremely important for practically exploiting and applying to real world cases what presented
3) the numerical experiments which have been performced are rich and well designed, while achived results are commented in a clear and effective manner.

**Q4 Main Weakness:**

I see few weacknesses in this paper.
In particular, the results of the performed numercial experiments do not clearly establish the proposed approch as superior when compared to existing methods from the specialized literature.
However, it is clear that the proposed approach achieves results which are comparable to those achieved by competing methods and no unique method appears to be winner under all numerical experiments settings, with specific reference to whatreported in Table 1 an din Table 2.
It is also clear that this could be normally expected for all machine learning algorithms.
I also would like to read more criticisms about your approach, I mean I would have appreciated to read which the weks point of the proposed approach from you.

**Q5 Detailed Comments To The Authors:**

I enjoyed reading your paper, which tackles a relevant problem, both theoretically and practically, and it is well structured and clearly written.
I have no major criticisms but I would like to know more on what your toughts concerning the results achieved by the proposed method as reported in Tabel 1 and in Table 2.
I also would like to know whether you think that testing robustness of your method with respect to how many sources are truly to be included into the linear combination. I mean, what happen when the number of sources increases from asmall number to a big one?
It may be that the target is the in covex hull of only some of the available sources, what happens to your method under this setting?

**Q7 Justification For Your Score:**

The paper tackles a theoretically and practically relevant problem and makes it in a clear, formal ancd convincing manner.
The paper is well structured and written and giudes the reader thtough the problem while clearly motivating the reasons of the proposed approach. the analysis of the specialized literature is appropriate and provides the interesred reader with an effective picture of other existing approaches to the same problem. Numerical experiments are well designed and perforrmed.

**Q9 Complying With Reviewing Instructions:**

1: Yes.

---

### Official Review · Reviewer_fJxh · 2022-04-12

**Q2(1) Originality/Novelty:** 3
**Q2(2) Significance/Impact:** 2
**Q2(3) Correctness/Technical Quality:** 3
**Q2(6) Clarity Of Writing:** 4
**Q6 Overall Score:** 7
**Q8 Confidence In Your Score:** 4

**Q1 Summary And Contributions:**

The paper proposes an extension of JDOT to multiple source domain adaptation.
The methods uses a weight for each source domain, that gets updated iteratively.
The paper includes experiments on standard DA datasets, with good results.

**Q2 Assessment Of The Paper:**

More detailed information regarding each of these aspects is given below:

**Q2(4) Quality Of Experiments (Optional):**

3: Good: The experimental evaluation is adequate, and the results convincingly support the main claims.

**Q2(5) Reproducibility:**

3: Good: Key resources (e.g., proofs, code, data) are available and key details (e.g., proofs, experimental setup) are sufficiently well-described for competent researchers to confidently reproduce the main results.

**Q3 Main Strengths:**

* The method looks like a clear improvement over other methods for multi source domain adaptation.
* The motivation makes sense.
* The experiments are done correctly, and described in enough detail

**Q4 Main Weakness:**

* The only comparison is with other JDOT-style methods.

**Q5 Detailed Comments To The Authors:**

* What about comparison with other DA methods, such as DANN (Ganin et al., 2016), RTN (Long et al. 2016), MDD (Zhang et al. 2019), etc. These are all single source algorithms, but they can be applied in the multi source setting in the same way as CJDOT.

* Table 2: The results from Montesuma and Mboula [2021] use the wrong table from that paper. They included both a Music
Genre Recognition (MGR) and a Music-Speech Discrimination (MSD) task in their table 2. This work is doing MSD, and should therefore compare to their MSD results.

* Lemma 1 seems trivial, and is not specific to convex combinations of source distributions at all

* Minor issue: Two different meanings of β are used: As the weight for |z-z'| in the equation for D, between (2) and (3), and as dual variables at the end of section 3.2.

* The appendix C2 seems to introduce new variants of the algorithm

* Surprisingly, Montesuma and Mboula [2021] also include results from an earlier version of this paper, under the name WJDOT.
(I found a reference to an arxiv version of this UAI2022 paper in their work. This looks like an opportunity for creating a cycle in the citation graph.). Why was the name changed to MSDA-WJDOT? Or is that a different algorithm?

* The name is a bit of a mouthful

**Q7 Justification For Your Score:**

The results are good.
The method is an elegant extension of earlier work.


**Q9 Complying With Reviewing Instructions:**

1: Yes.

---

### Official Review · Reviewer_nJEp · 2022-04-13

**Q2(1) Originality/Novelty:** 3
**Q2(2) Significance/Impact:** 3
**Q2(3) Correctness/Technical Quality:** 3
**Q2(6) Clarity Of Writing:** 3
**Q6 Overall Score:** 6
**Q8 Confidence In Your Score:** 3

**Q1 Summary And Contributions:**

Authors address the problem of domain adaptation from a new perspective: instead of crushing diversity of the source distributions, we exploit it to adapt better to the target distribution. It aims at simultaneously finding an Optimal Transport-based alignment between the source and target distributions and a re-weighting of the source distributions. Authors give both theoretical and empirical results.

**Q2 Assessment Of The Paper:**

More detailed information regarding each of these aspects is given below:

**Q2(4) Quality Of Experiments (Optional):**

3: Good: The experimental evaluation is adequate, and the results convincingly support the main claims.

**Q2(5) Reproducibility:**

2: Fair: Key resources (e.g., proofs, code, data) are unavailable but key details (e.g., proof sketches, experimental setup) are sufficiently well-described for an expert to confidently reproduce the main results.

**Q3 Main Strengths:**

1) Authors clearly point out the differences between their proposed method and existing methods.
2) Proposed solution has a clear motivation and authors back their claims using both theoretical and empirical results


**Q4 Main Weakness:**

1) Missing literature - Some of the related problems like domain generalization should briefly be touched and some methods that use distribution similarity or optimal transport in the domain generalization literature should be included.

2) Some missing details like tuning and time complexity discussions


**Q5 Detailed Comments To The Authors:**

1) Missing literature - Some of the related problems like domain generalization should briefly be touched and some methods that use distribution similarity or optimal transport in the domain generalization literature should be included in the related work. For example, Blanchard et al (2021) use a similarity between distributions to address the issue. One may use the optimal transport type of metric to get these distances/similarity between distributions.

2) “how to perform the validation of the parameters including early stopping. We propose here to use the sum of squared errors (SSE) between the target points in the embedding and their cluster centroids”: Is this well accepted in the literature? If yes, would it be possible to cite it? How were baselines tuned?

3) In Table 1, why is Target performing so much worse in the DSLR method?

4) One thing that was missing from the discussion was the time complexity of the proposed method and other baselines. Can authors give some information about how long was the training/optimization for experimental results in the paper?

5) “ Interestingly MSDA-WJDOT can even outperform Target due to its access to a larger number of sample” and “Since we have access to labels for Target and Baseline+Target, we validate the model by using the classification accuracy on the target validation set making those two approaches clear upper bounds on the attainable performance for each dataset”: These two statements are contradictory. Can authors clarify these?

[1] Blanchard, Gilles, Aniket Anand Deshmukh, Urun Dogan, Gyemin Lee, and Clayton Scott. "Domain Generalization by Marginal Transfer Learning." Journal of Machine Learning Research 22 (2021): 1-55.


**Q7 Justification For Your Score:**

1) Missing literature - Some of the related problems like domain generalization should briefly be touched and some methods that use distribution similarity or optimal transport in the domain generalization literature should be included.

2) Some missing details like tuning and time complexity discussions


**Q9 Complying With Reviewing Instructions:**

1: Yes.

---

### Official Review · Reviewer_DFbu · 2022-04-13

**Q2(1) Originality/Novelty:** 3
**Q2(2) Significance/Impact:** 2
**Q2(3) Correctness/Technical Quality:** 3
**Q2(6) Clarity Of Writing:** 3
**Q6 Overall Score:** 5
**Q8 Confidence In Your Score:** 3

**Q1 Summary And Contributions:**

The paper proposes MSDA-WJDOT for domain adaptation from multiple sources to target distribution. They present theoretical bound for the error and an algorithm to solve the problem. Experiments on synthetic and real datasets are given to justify their method.

**Q2 Assessment Of The Paper:**

More detailed information regarding each of these aspects is given below:

**Q2(4) Quality Of Experiments (Optional):**

3: Good: The experimental evaluation is adequate, and the results convincingly support the main claims.

**Q2(5) Reproducibility:**

3: Good: Key resources (e.g., proofs, code, data) are available and key details (e.g., proofs, experimental setup) are sufficiently well-described for competent researchers to confidently reproduce the main results.

**Q3 Main Strengths:**

The paper extends the method and theories from [1] below to the case of multiple sources. It is nice to have theories for the generalized case of multiple sources. Experiments are also good, validating their method's practical performance.


[1] Nicolas Courty, Rémi Flamary, Amaury Habrard, and Alain Rakotomamonjy. Joint distribution optimal transportation for domain adaptation. In Advances in Neural Informa- tion Processing Systems 30, pages 3730–3739. 2017.E

**Q4 Main Weakness:**

This work is too incremental from [1]. They do compensate it with careful experiments. However, more theoretical or methodological contributions would strengthen the paper.

**Q5 Detailed Comments To The Authors:**

I think settings like Federated Learning would benefit most from multiple source domain adaptation considered in this paper. The authors can consider developing algorithm that solves their problem MSDA-WJDOT in FL setting. This would also better motivate their problem.

**Q7 Justification For Your Score:**

Though incremental, this work extending known result to the case of multiple sources domain adaptation has its own merit. Especially the experiments are well carried out. However, the authors can use more well-known real datasets and should extend theories more to strengthen the paper.

**Q9 Complying With Reviewing Instructions:**

1: Yes.

---

### Decision · Program_Chairs · 2022-05-15

**Decision:**

Accept (Poster)

**Comment:**

Meta Review: This paper proposes a method for multi-source domain adaptation. The basic ideas is aims to find simultaneously an Optimal Transport-based alignment between the source and target distributions and a re-weighting of the sources distributions.  The method is an elegant extension of previous work, with interesting theoretical advances and convincing empirical results.